# Cellular and Extracellular Vesicle RNA Analysis in the Global Threat Fungus *Candida auris*

Isadora F. Munhoz da Rocha,[a] Sharon T. Martins,[a] Rafaela F. Amatuzzi,[a] Daniel Zamith-Miranda,[b,c] Joshua D. Nosanchuk,[b,c] Marcio L. Rodrigues,[a,d] Lysangela R. Alves[a]

[a]Gene Expression Regulation Laboratory, Carlos Chagas Institute, Fiocruz Paraná, Curitiba, Brazil
[b]Department of Microbiology and Immunology, Albert Einstein College of Medicine, Bronx, New York, USA
[c]Division of Infectious Diseases, Department of Medicine, Albert Einstein College of Medicine, Bronx, New York, USA
[d]Microbiology Institute, Federal University of Rio de Janeiro, Rio de Janeiro, Brazil

**ABSTRACT** Emerging and reemerging pathogens are a worldwide concern, and it is predicted that these microbes will cause severe outbreaks. *Candida auris* affects people with weakened immune systems, particularly those who are hospitalized or are in health care facilities. Extracellular vesicles (EVs) are lipid bilayer structures released by organisms from all domains of life. EVs can deliver functional molecules to target cells, including proteins and nucleic acids, especially RNA molecules. EVs from several pathogenic fungi species play diverse biological roles related to cell-cell communication and pathogen-host interaction. In this study, we describe a data set which we produced by sequencing the RNA content of EVs from *C. auris* under normal growth conditions and in the presence of the antifungal caspofungin, a first-line drug to treat this fungus. To generate a more complete data set for future comparative studies, we also sequenced the RNA cellular content of EVs under the same conditions. This data set addresses a previously unexplored area of fungal biology regarding cellular small RNA and EV RNA. Our data will provide a molecular basis for the study of the aspects associated with antifungal treatment, gene expression response, and EV composition in *C. auris*. These data will also allow the exploration of small RNA content in the fungal kingdom and might serve as an informative basis for studies on the mechanisms by which molecules are directed to fungal EVs.

**IMPORTANCE** *Candida auris*, a relevant emerging human-pathogenic yeast, is the first fungus to be called a global public health threat by the WHO. This is because of its rapid spread on all inhabited continents, together with its extremely high frequency of drug and multidrug resistance. In our study, we generated a large data set for 3 distinct strains of *C. auris* and obtained cellular small RNA fraction as well as extracellular vesicle RNA (EV-RNA) during normal growth conditions and after treatment with caspofungin, the first-line drug used to treat *C. auris* infection.

**KEYWORDS** *Candida auris*, RNA-seq, dataset, extracellular vesicles, small RNA

Invasive fungal infections are a serious health problem worldwide, and pathogens belonging to the genus *Candida*, mainly non-*albicans* species, are responsible for more than 50% of all blood infections (1). Immunocompromised individuals, patients affected by major surgeries, patients admitted to an intensive care unit (ICU), or patients in long-term-care facilities, are more susceptible to disseminated candidiasis (1), of which more than 45% of cases can lead to death (2).

 *C. auris* was identified in 2009 in Japan and rapidly spread globally, causing several lethal outbreaks in hospitals (3). Pan-resistance occurs in ~10% of clinical isolates; more than half are multidrug-resistant (4, 5) and about 90% of the strains are resistant to fluconazole (6, 7). Due to its intrinsic resistance to antifungals and its high lethality

Address correspondence to Lysangela R. Alves, Lysangela.alves@fiocruz.br.

in critically ill patients, *C. auris* was the first fungus to be considered a global threat by the USA Centers for Disease Control (CDC).

Despite the increasing medical importance of *C. auris*, little is known about its mechanisms of pathogenicity and virulence (8). The *C. auris* genome includes many factors potentially affecting disease-related processes, including proteinases, lipases, phospholipases, adhesins, efflux pumps, regulators of biofilm formation, and drug transporters involved with azole resistance (9). Fungi secrete a myriad of immunoreactive molecules, cytotoxic proteins, virulence factors, and a subset of functional RNAs, which assist them both in pathogenic processes and in the environment (10, 11). Several studies have revealed that extracellular vesicles (EVs) are vehicles participating in the delivery of these molecules into the outer milieu (12, 13). EVs are lipid bilayer structures released by organisms from all domains of life (14, 15). EVs produced by pathogenic fungi play diverse biological roles related to cell-cell communication and pathogen-host interaction (16–20). There is an association between fungal EVs and the antifungal response (21). However, the mechanisms behind this association are not known. A variety of RNA molecules are present in fungal EVs, including noncoding RNAs (ncRNAs), mRNAs, nucleolar RNAs (snoRNAs), small nuclear RNAs (snRNAs), tRNA fragments, microRNA (miRNA)-like, and antisense RNAs (asRNAs) (22–24). The diversity in the composition of EVs suggests multiple effects on the interaction between pathogens and their hosts (24).

Regulatory ncRNAs include miRNAs, small interfering RNAs (siRNAs), circular RNAs (circRNAs), asRNAs, and long ncRNAs (lncRNAs) (25). They play important roles in many aspects of RNA metabolism, including splicing, transcription, translation, and chromatin remodeling (25, 26). Despite being extensively studied in animals and plants, ncRNAs are poorly understood in fungi. One important pathway that generates small regulatory RNAs is the RNA interference pathway. This pathway is comprised of the Dicer protein, an RNase III that generates small interfering RNA molecules (siRNAs), and the Argonaute protein (Ago), which associates with the siRNAs and leads to translational or transcriptional repression of the target RNAs (27). Some fungal species possess RNA interference pathways similar to those of higher eukaryotes; however, some proteins of the RNA interference pathway are absent in a significant number of fungal species. Nevertheless, many of these organisms can produce small RNA molecules resembling siRNAs, which originate from distinct pathways, generating a great diversity of small regulatory RNA molecules (28, 29). In *Saccharomyces cerevisiae*, anti-sense RNAs play a role in transcription regulation. For instance, the PHO system responds to nutritional alterations in the cell, as inferred from models of inorganic phosphate (Pi) starvation. The transcriptional regulator Pho4 activates the transcription of intragenic and antisense RNA molecules, leading to the downregulation of complementary mRNAs (30). The lncRNAs are associated with important roles in gene regulation and disease. RNA-seq data from *Schizosaccharomyces pombe* have demonstrated the presence of 5,775 novel lncRNAs, many of which are associated with the cell cycle (31). In *S. cerevisiae*, asRNAs can repress the expression of the associated mRNAs (32).

Despite the abundance of cellular and extracellular small RNA molecules in fungi, they have been poorly explored. In addition, there is little availability of large data sets for sRNA analysis in these organisms. For this scenario, an overview of our study design is illustrated in Fig. 1. To the best of our knowledge, this is the first study to associate small RNA data sequencing in the cell with the RNAs present in the extracellular space. Furthermore, little is known about either the regulation of gene expression in *C. auris* or the mechanisms linked to intrinsic multiresistance. The role of EVs in this process is also unknown. In this work, we generated a descriptive data set providing detailed information on cellular small RNA content and EV-associated RNA. In addition, we report on the molecular response to the antifungal caspofungin, expanding information on how this compound affects *C. auris*. Finally, these data can provide useful material for the study of small RNA content in the fungal kingdom, and for the exploration of the mechanisms that direct molecules to EVs.

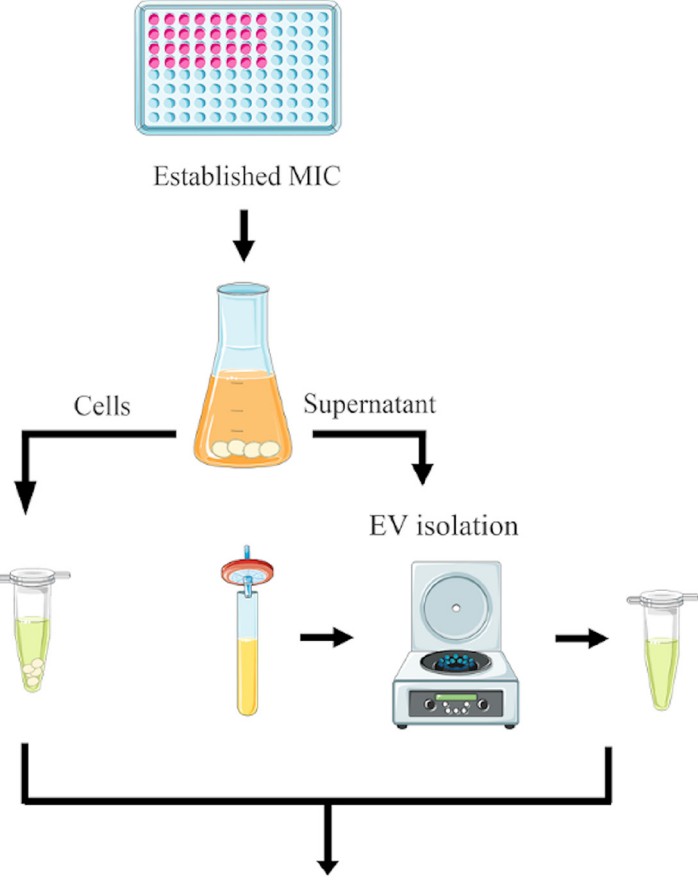

Established MIC

Cells

Supernatant

EV isolation

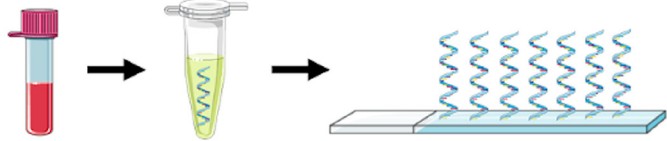

RNA library preparation and sequencing

**FIG 1** Schematic presentation of yeast cells and supernatant preparations for EV isolation followed by RNA-seq data generation.

## RESULTS

***C. auris* growth upon caspofungin treatment and EV isolation.** To characterize the RNA composition of *C. auris* EVs and to further compare it with the cellular RNA content, we extracted and sequenced RNA molecules from both fungal cells and EVs. To ensure the accuracy of the cell-EV compositional comparisons, we used three distinct strains belonging to the clades I (B8441 and MMC1) and III (B11244). In addition, we also evaluated EV production and composition in response to caspofungin. For this, *C. auris* cells were grown in the presence of subinhibitory concentrations of caspofungin as previously described (21), and EVs were isolated from the culture supernatants (21, 33).

The MICs corresponded to 0.25 $\mu$g/mL (B8441), 0.5 $\mu$g/mL (B11244), and 2 $\mu$g/mL (MMC1) (33, 34). For the RNA-seq data, we determined the sub-inhibitory concentrations of caspofungin (as described in the methods section), allowing for the analysis of the drug response in living cells as previously described (21). The cells were collected for RNA extraction as described in the Methods section while the supernatant was processed for EV isolation. The EV concentration and size were analyzed by nanoparticle tracking analysis (NTA) and transmission electron microscopy. The EVs displayed

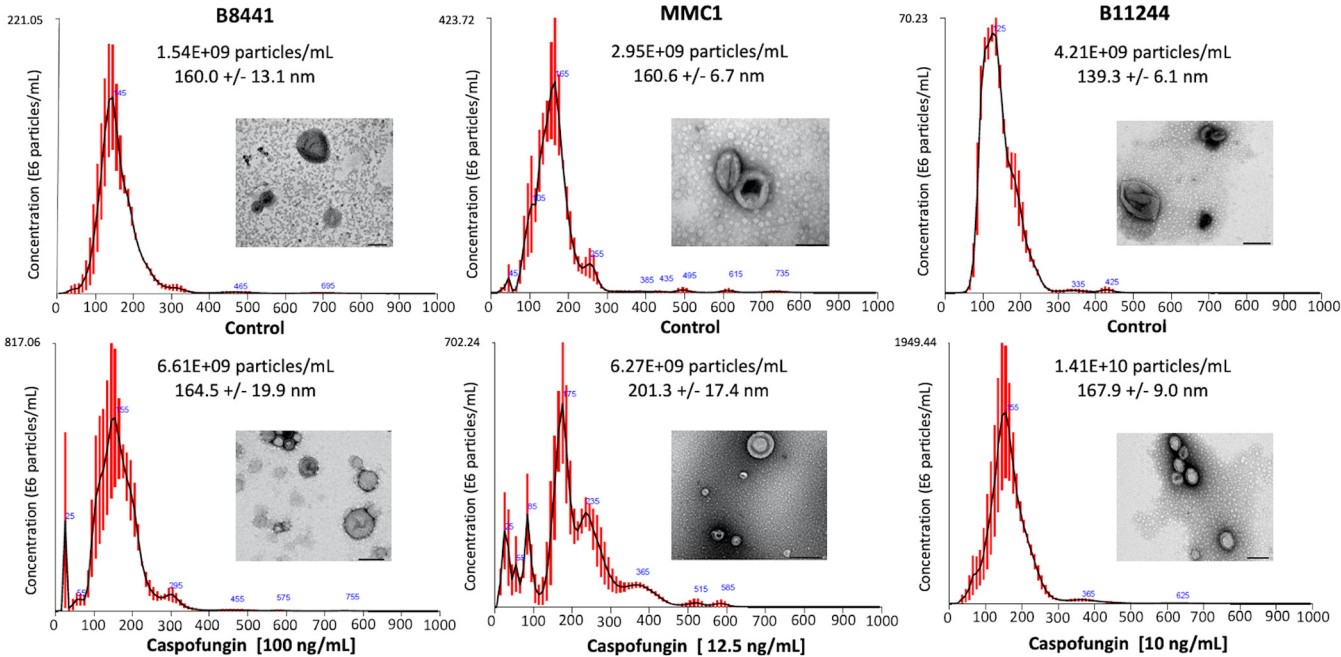

**FIG 2** Characterization of *C. auris* EVs. Extracellular vesicle size distribution graphs for B8441, B11244 and MMC1 EVs. Average sizes are indicated in nm. Transmission electron microscopy images of EVs isolated from control and caspofungin treatment groups are shown for each strain. Bars = 100 nm.

the standard morphology of "cup-shaped" particles with lipid bilayers under both control and caspofungin-treated conditions (Fig. 2). The EV concentration was higher in cultures subjected to caspofungin treatment (Fig. 2).

**RNA and cDNA sequencing quality control.** The isolated RNAs from the cells and the EVs were quantified by fluorometry (Qubit V2, Invitrogen), and on a 2100 Agilent Bioanalyzer (Agilent). We also assessed the RNA integrity number (RIN) of the cellular samples used for the small RNA library preparation using the Eukaryote Total RNA Pico Chip (Fig. 3). A summary of the cell numbers and RNA concentrations is listed in Table 1. The 18 and 28S rRNAs peaks as well as the small RNA fraction were observed. Overall, the RIN of the samples varied from 7 to 8.9, presenting therefore high integrity, and they were used for library construction. The samples were prepared with the TruSeq small RNA Sample Preparation kit, and the cDNAs obtained were analyzed to determine quality and quantity using an Agilent 2100 Bioanalyzer (Fig. 4).

**Sample analysis.** Cellular and EV RNAs were sequenced in experimental triplicates for each strain as well as for control conditions or after caspofungin treatment. The small RNA-seq data were deposited in the SRA repository (NCBI). After sequencing 55 bp, the reads were trimmed to remove the internal adapter and most of the sequences varied from 50 to 55 nt (Fig. 5A). Next, we evaluated the quality of the sequences generated. Overall, the reads presented a Q30 or higher value of quality score (Fig. 5B). To evaluate the reproducibility of the samples, they were grouped according to their origin, condition, and strain (cell versus EV, caspofungin versus control) using principal-component analysis (PCA) (Fig. 6). In addition, the samples were mapped against the *C. auris* genome, and the mapping statistics are listed in Table 2. Mapping was performed using two databases: one with the genomic information to map against the mRNAs and intergenic regions, and the ncRNA database, as the small RNA fractions were enriched for these molecules (Table 2).

In order to corroborate the RNA-seq data, real-time quantitative PCR (qPCR) was applied to the strains under the control condition and with caspofungin treatment. We arbitrary selected 6 transcripts with variable expression levels in EVs and in cells. Based on the expression levels from the RNA-seq described as TPM (transcripts per million), and the comparison with the quantitative real-time PCR, it was possible to observe a correlation in mRNA levels between the different techniques (Fig. 7).

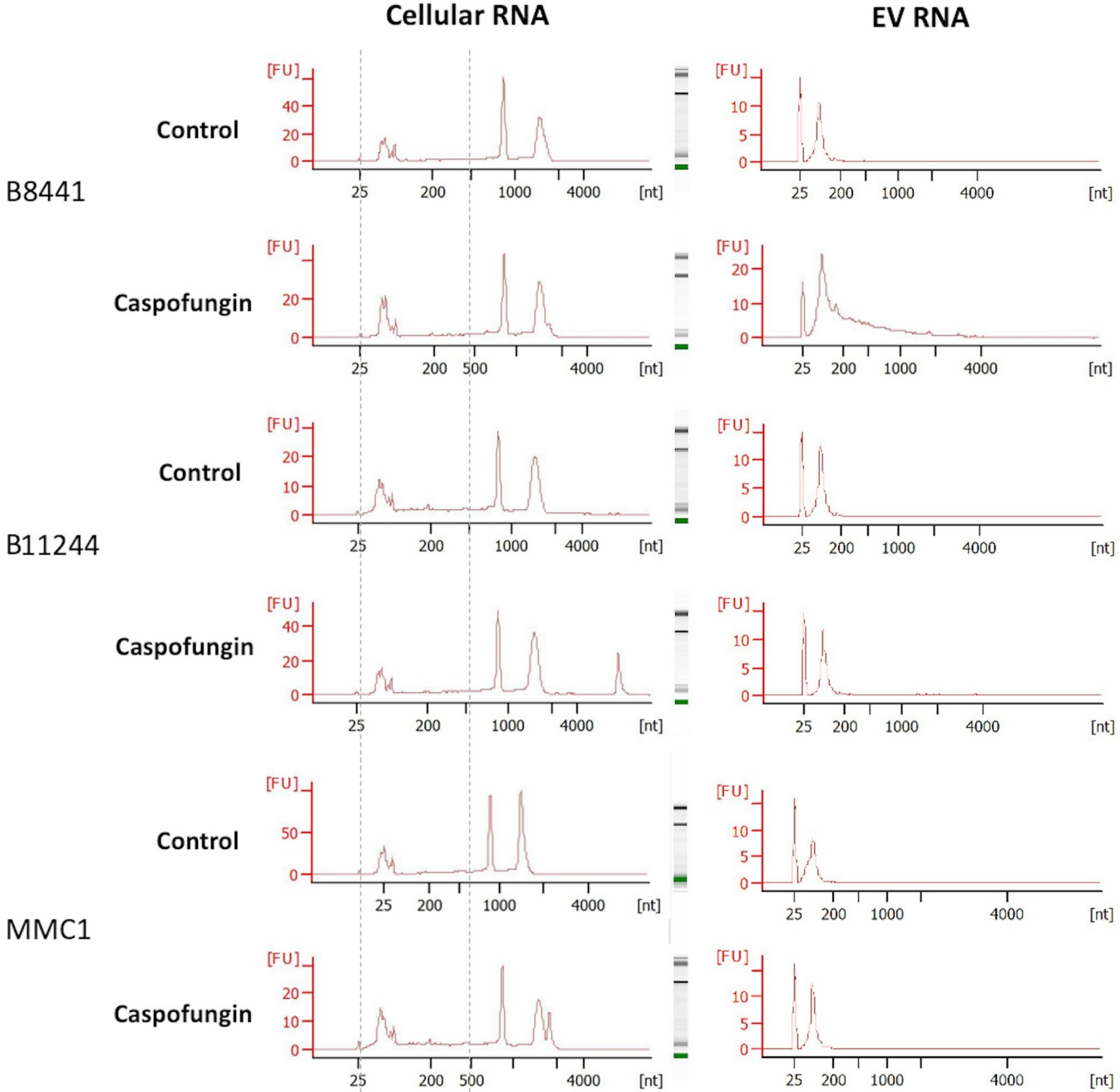

**FIG 3** RNA profile from cellular and EV samples. Representative images correspond to one of the replicates used in the study for each strain for both control and caspofungin treatment groups. The *x* axis indicates size in nucleotides (nt) and the *y* axis indicates fluorescence units (FU). The peak at 25 nt indicates the internal marker of the run. Samples were analyzed in a Eukaryote Total RNA Nano chip. Gray dotted lines represent the region corresponding to the cellular small RNA fraction.

## DISCUSSION

Caspofungin treatment led to alterations in the *C. auris* cell morphology (21) and aberrant cell phenotypes have been observed in other studies with *Candida* species (35, 36), including *C. auris*, when the cells were treated with a 1,3-$\beta$-d-glucan synthase inhibitor (37). In addition, it was observed that caspofungin treatment led to increased EV production in *S. cerevisiae* (38). These premises initially led us to investigate the response of *C. auris* to caspofungin, with a focus on RNA composition and EV production.

EVs are important tools used by fungal cells to export RNA (12, 22, 39). However, the pathways and mechanisms that select molecules to be directed to the EVs are unknown to

**TABLE 1** RNA concentrations of the samples used in this study

| Cellular RNA | | Extracellular vesicle RNA | | | |
|---|---|---|---|---|---|
| Sample | Total RNA concn ($\mu$g) | No. cells $\times$ $10^6$/mL | No. of cells/ expt | EV RNA/400 mL of culture (ng) | Normalized EV RNA/$10^8$ cells (pg) |
| B8441 | | | | | |
| 1 | 1.5 | 3,875 | 1.55E+12 | 20.2 | 13 |
| 2 | 1.45 | 5,400 | 2.16E+12 | 36.7 | 17 |
| 3 | 1.38 | 2,350 | 9.40E+11 | 11.5 | 12.2 |
| Caspo 1 | 1.09 | 900 | 3.60E+11 | 252 | 700 |
| Caspo 2 | 0.99 | 800 | 3.20E+11 | 340.8 | 1,065 |
| Caspo 3 | 0.77 | 640 | 2.56E+11 | 148.3 | 579.4 |
| | | | | | |
| MMC1 | | | | | |
| 1 | 1.48 | 1,875 | 7.50E+11 | 13 | 17.3 |
| 2 | 1.54 | 2,850 | 1.14E+12 | 26.2 | 22.9 |
| 3 | 1.52 | 2,625 | 1.05E+12 | 16.2 | 15.4 |
| Caspo 1 | 1.62 | 3,350 | 1.34E+12 | 42 | 31.3 |
| Caspo 2 | 1.3 | 3,000 | 1.20E+12 | 24.7 | 20.6 |
| Caspo 3 | 1.39 | 5,125 | 2.05E+12 | 77.3 | 37.7 |
| | | | | | |
| B11244 | | | | | |
| 1 | 1.2 | 4,075 | 1.63E+12 | 12.1 | 7.4 |
| 2 | 1.27 | 3,175 | 1.27E+12 | 19.4 | 15.3 |
| 3 | 1.03 | 2,075 | 8.30E+11 | 4.7 | 5.6 |
| Caspo 1 | 0.98 | Large | Large clumps | 11.1 | large clumps |
| Caspo 2 | 1.36 | Large | Large clumps | 27.8 | large clumps |
| Caspo 3 | 0.84 | Large | Large clumps | 10.5 | large clumps |

date. Despite this, EV RNAs play significant roles in fungal biology, including interstrain virulence transfer in *Cryptococcus gattii* (12). This observation proves the concept that EV-mediated RNA export participates in cell-to-cell communication in fungi (12).

Most of the RNA molecules in fungal EVs are small RNAs from distinct classes, including small-nucleolar RNAs (snoRNAs), small nuclear RNA (snRNAs) and tRNA fragments (tfRNAs) (39). Other regulatory small RNAs were identified in the EVs. For instance, *Malassezia sympodialis* EVs contain 16- to 22-nt small RNAs, similar to miRNA molecules (19). In *Pichia fermentans*, miRNA-like molecules present in the EVs were associated with the dimorphic transition from yeast-like to pseudohyphal morphology (40). *Histoplasma capsulatum* EVs contained 25-nt-long anti-sense RNAs that could play a role in the RNA interference machinery (41). In addition, a few studies have indicated that the EV-RNA composition differs from the cellular composition (41, 42). As an example, rRNA, which comprises more than 90% of cellular RNA, was barely detected in the fungal EVs (41, 42). This observation reinforces the importance of comparing cellular and EV RNAs, since such analysis could reveal particularities in the traffic of nucleic acids in fungal cells.

In summary, this is the first study that associates data on small RNA sequencing in the cell with the RNAs present in EVs in fungi. In addition, little is known about the regulation of gene expression in *C. auris*, the mechanisms linked to intrinsic multiresistance, and the role of EVs within this process. Our data set provides useful information about the yeast cell small RNA content in cells and EVs, as well as small RNA content in the presence of caspofungin, and can be used to investigate antifungal response and targets for therapy and treatment. These data can be used to develop new information about *C. auris* biology and ncRNAs in fungi in general, an area that needs to be better explored to improve our capacity for preventing and treating global infectious threats.

## MATERIALS AND METHODS

**Fungal growth conditions.** *C. auris* B8441, B11244, and MMC1 stains were maintained at −80°C. The B8441 and B11244 strains were obtained from the CDC and the MMC1 strain was isolated from a

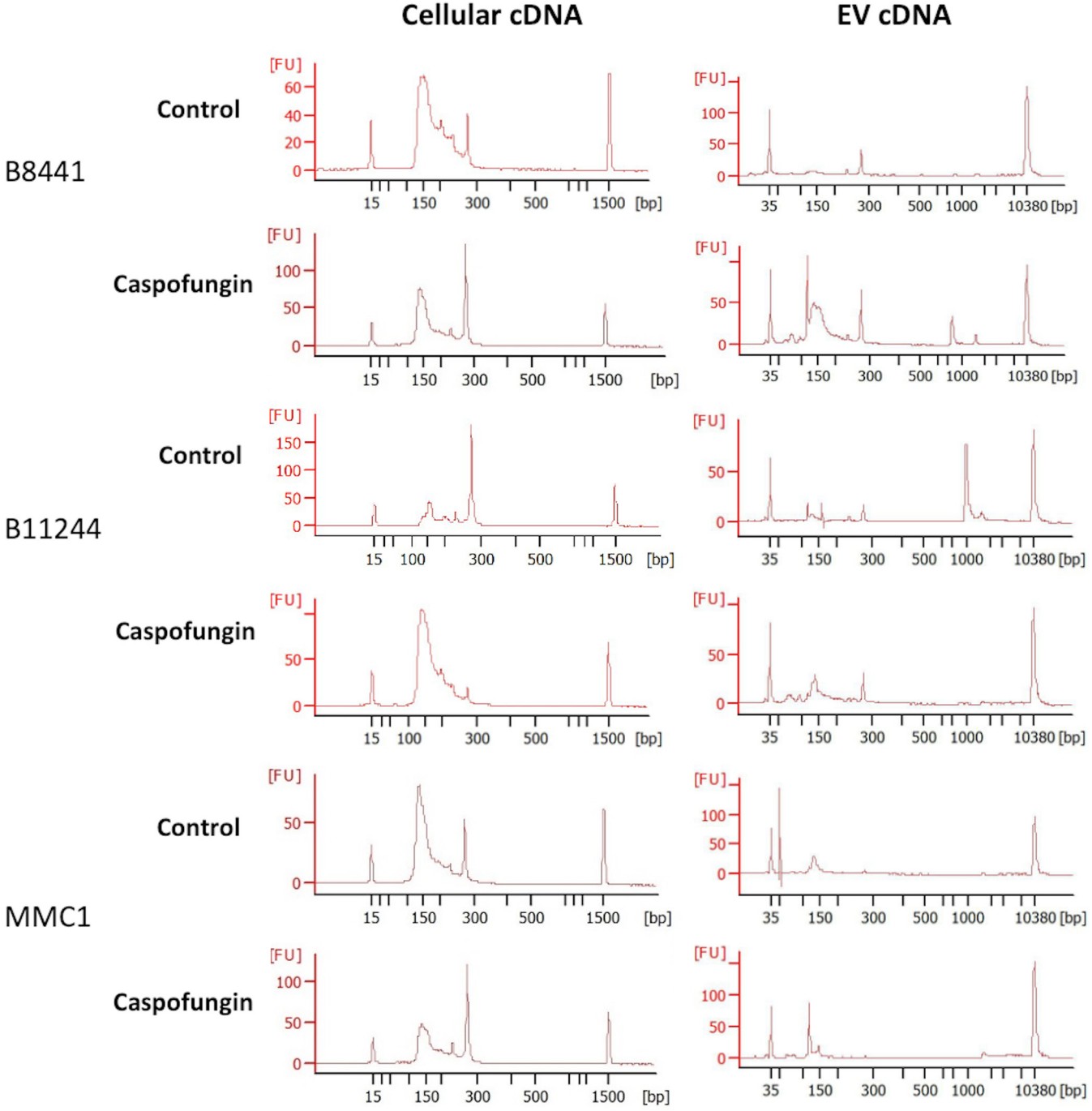

**FIG 4** cDNA profile from cellular and EV samples. Each representative images corresponds to one of the replicates used for each strain for both control and caspofungin treatment groups. The *x* axis indicates size in base pairs (bp) and the *y* axis indicates fluorescence units (FU). For the cellular cDNA, samples were analyzed in a DNA 100, and the peaks at 15 and 1,500 bp indicate the internal markers of the run. For the EV cDNA, samples were analyzed in a high-sensitivity DNA assay and the peaks correspond to 35 and 10,380 bp.

patient in Montefiore Medical Center in New York (33). After thawing in Sabouraud broth, suspensions were incubated at 30°C for 24 h. Yeast cell suspensions were then plated onto Sabouraud agar plates and incubated at 30°C for 48 h. The plates were then stored at 4°C (for no longer than 4 weeks) and used in experiments. Based on the MIC, growth curves of *C. auris* were performed with or without the presence of caspofungin (Sigma-Aldrich) at subinhibitory concentrations of 100 ng/mL (B8441), 10 ng/mL (B11244), or 12.5 ng/mL (MMC1). The assays were performed in technical and biological triplicates in a 96-well flat-bottomed translucent plate to a final volume of 200 $\mu$l per well and $5 \times 10^5$ cells/mL. The cells were incubated in a microplate reader (Synergy Biotek) for 72h at 30°C, and optical density (OD) readings were taken every 1 h with a wavelength of 540 nm, with shaking of the plate for 30 s prior to

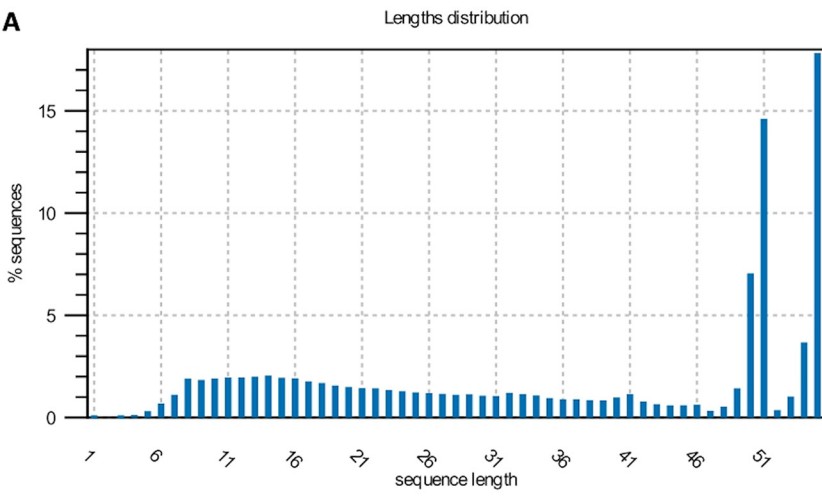

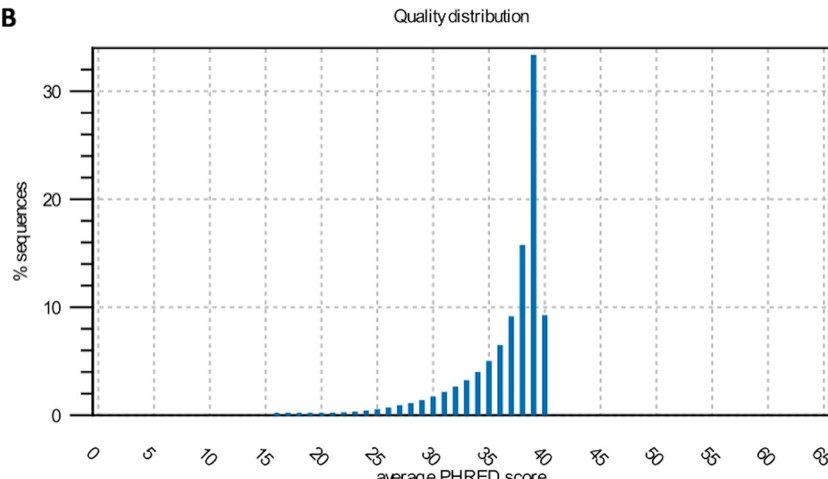

**FIG 5** Sequencing reads size and quality control. (A) Sequence length distribution obtained after trimming. The *x* axis corresponds to sequence length and the *y* axis correspond to the percentage (%) of sequences distributed in respective length. (B) Quality distribution of the reads based on the Q-value represented as the average PHRED score.

each reading. A nonlinear model, the Gompertz curve, was used to analyze growth curves, and significance was calculated by analysis of variance (ANOVA) comparing control versus treatment. The results represent the mean $\pm$ standard deviation of three independent experiments. The curves and analyzes were performed using the GraphPad Prism 8.0 program.

**EV isolation.** One colony from each strain was inoculated into Sabouraud broth and incubated for 24 h at 30°C, with shaking at 200 rpm. The cell density was adjusted to $10^6$ cells/mL in a total volume of 400 mL, with or without the addition of caspofungin, under the same conditions as described for fungal growth. After 24 h at 30°C (200 rpm), the cells were pelleted (8,000 rpm, 15 min at 4°C) and the supernatant was centrifuged at 14,000 rpm for 15 min at 4°C. Then, the resulting supernatant was filtered through a 0.45-$\mu$m membrane (Millipore) followed by concentration on an Amicon ultrafiltration system with a 100-kDa cutoff. The concentrated supernatant was ultracentrifuged for 1 h at 150,000g at 4°C, washed with phosphate-buffered saline (PBS), and ultracentrifuged again under similar conditions. The supernatant was discarded and the resulting EVs were suspended in 100 $\mu$L of PBS.

**Nanoparticle tracking analysis.** NTA was used to determine the concentration and size of EVs isolated from *C. auris*, using a Nanosight LM-10 (Malvern Panalytical). EV samples were diluted in filtered PBS prior to injection, and the dilution factor used varied from 1:10 to 1:50 so that reading concentrations could be within the ideal range of $1 \times 10^8$ to $1 \times 10^9$ particles/mL, as recommended (43). The injection using a syringe pump was set to 20 followed by image capture. The videos were set to three runs of 60 s each, with the detection threshold defined as 2 and the camera level as 9. The data were analyzed using NTA 3.1 software. For each experimental group, we evaluated the EV samples in triplicates: 5 videos of 60 s each were captured for each sample, with 1,500 frames per video file and at least 20 particles per frame (43). Total yield (EV particles/mL) was calculated based on dilution factors. Statistical analysis was performed with Minitab Statistical Software 17.0, where samples were submitted to one-way ANOVA and values were compared with a Tukey test with 5% probability. The samples were prepared in biological triplicates with five technical replicates.

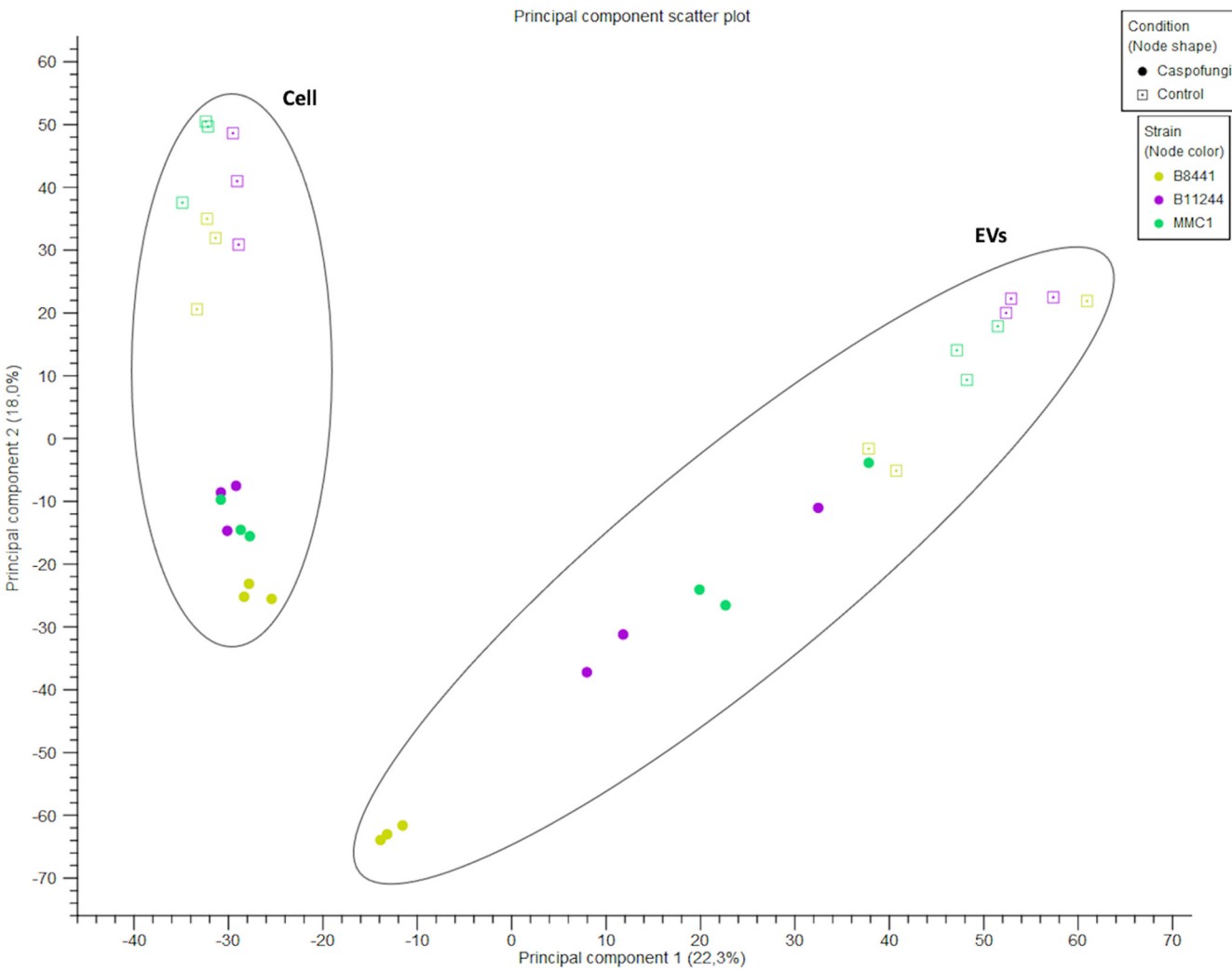

**FIG 6** RNA-seq experiment quality control. Principal-component analysis of the 36 samples sequenced in this study.

**RNA isolation and sequencing.** Total RNA along with the small RNA fraction from the yeast cells ($1 \times 10^7$) was isolated using a miRNeasy kit (Qiagen) with modifications. A 1:1 volume of glass beads was added to the TRIzol buffer, followed by 10 rounds of vortex agitation (each round, 1 min at 4°C) in order to disrupt the cell wall. After centrifugation, small RNAs were collected according to the manufacturer's instructions. For the EV, the RNA was isolated with the miRNeasy kit (Qiagen) according to the manufacturer's instructions. The DNA cleanup step was performed with all samples using the RNase-free DNase protocol (Qiagen). For RNA quantification and integrity analysis, we used a Qubit fluorometer (Thermo Fisher) and an Agilent 2100 Bioanalyzer (Agilent Technologies). For the cellular small RNA and the EV RNA, the libraries were constructed with 100 ng of RNA (cell and EV) using the TruSeq small RNA kit (Illumina). Libraries were prepared according to the manufacturer's instructions with slight adaptations. For the step of size-selecting the samples by using the 6% acrylamide gel, instead of cutting between custom markers, we cut the molecular sizes corresponding to 140- to 500-nt long (Fig. 2). The samples were prepared in three independent replicates and RNAseq was performed on a HiSeq 2500 (Illumina, single-end 50-bp SR mid output run) at the Life Sciences Core Facility (LaCTAD), at the University of Campinas (UNICAMP).

**RNA-seq data analysis.** We sequenced 36 samples in total, and the sequences in fastq format were analyzed by CLC Genomics Workbench v 20.0.03 (Qiagen). The first analysis performed was the adapter trimming to remove the internal adapter that might have been present in the shorter sequences. The TruSeq small RNA adapter sequence was TGGAATTCTCGGGTGCCAAGG. When the adapter was identified in the sequence, it was removed along with the following sequence (3' trim). The sequences were first mapped using the corresponding *C. auris* genome for strain B8441 (GCA_002759435.2 V2). For the ncRNA, we used the ncRNA from *Candida* genome databases and the reference available at the RNA central database: C_auris_B8441_version_sXX-mYY-rZZ_other_features_no_introns.fasta.gz and candida_auris_gca_001189475. ASM118947v1.ncrna.1. The use of two distinct references was to improve the ncRNA annotation, so that it could be present in one reference and absent in the other. The following parameters were used for the

**TABLE 2** Detailed summary of mapping statistics[a]

| Sample name | Adapter trimming (in nt) | | Map (%) | Mapped to (%) | | |
|---|---|---|---|---|---|---|
| | Before | After | | Genes | Intergenic | ncRNAs |
| B8441 | | | | | | |
| EVs 1 | 8264066 | 7538329 | 96.02 | 11.33 | 23.81 | 64.86 |
| EVs 2 | 4118186 | 740227 | 98.45 | 12.27 | 13.15 | 74.58 |
| EVs 3 | 3883453 | 373903 | 96.12 | 18.77 | 12.16 | 69.07 |
| Caspo EVs 1 | 6276373 | 6031814 | 98.33 | 5.24 | 53.95 | 40.81 |
| Caspo EVs 2 | 5628093 | 5309993 | 96.37 | 5.14 | 50.89 | 43.97 |
| Caspo EVs 3 | 5719892 | 5572477 | 97.63 | 4.76 | 53.13 | 42.11 |
| B11244 | | | | | | |
| EVs 1 | 4846009 | 4749427 | 96.87 | 54.38 | 34 | 11.62 |
| EVs 2 | 6023630 | 5134744 | 97.54 | 29.59 | 64.66 | 5.75 |
| EVs 3 | 4885722 | 4491008 | 97.07 | 56.18 | 30.44 | 13.38 |
| Caspo EVs 1 | 4928155 | 4827333 | 93.88 | 12.88 | 64.8 | 22.32 |
| Caspo EVs 2 | 6013768 | 8408963 | 95.69 | 11.39 | 61.77 | 26.84 |
| Caspo EVs 3 | 2118126 | 5902897 | 97.22 | 17.59 | 61.84 | 20.57 |
| MMC1 | | | | | | |
| EVs 1 | 5053359 | 3102092 | 94.03 | 11.76 | 36.3 | 51.94 |
| EVs 2 | 5424052 | 1201170 | 90.30 | 20.95 | 23.46 | 56.05 |
| EVs 3 | 5296279 | 4115195 | 96.41 | 12.2 | 20.14 | 67.66 |
| Caspo EVs 1 | 5117480 | 4666909 | 97.56 | 6.61 | 60.1 | 33.29 |
| Caspo EVs 2 | 10044543 | 5302577 | 96.99 | 9.49 | 54.44 | 36.07 |
| Caspo EVs 3 | 6301634 | 2024761 | 98.61 | 8.2 | 62.82 | 28.98 |
| B8441 sRNA | | | | | | |
| Cell 1 | 2869018 | 2866412 | 89.60 | 17.61 | 50.56 | 31.83 |
| Cell 2 | 2666340 | 2634661 | 93.55 | 11.22 | 63.95 | 24.83 |
| Cell 3 | 2835343 | 2833096 | 94.78 | 7.01 | 78.01 | 14.98 |
| Cell Caspo EVs 1 | 1688349 | 1687523 | 98.67 | 6.91 | 66.17 | 26.92 |
| sRNA cell Caspo EVs 2 | 2538767 | 2536100 | 97.14 | 6.71 | 63.14 | 30.15 |
| Cell Caspo EVs 3 | 1637451 | 1636479 | 96.61 | 7.53 | 63.18 | 29.29 |
| B11244 sRNA | | | | | | |
| Cell 1 | 2476330 | 2474720 | 94.84 | 11.09 | 19.76 | 69.15 |
| Cell 2 | 1694024 | 1671635 | 92.73 | 7.94 | 19.06 | 73.00 |
| Cell 3 | 2635661 | 2623670 | 98.45 | 4.35 | 22.53 | 73.12 |
| Cell Caspo EVs 1 | 1602934 | 1601776 | 97.32 | 6.54 | 64.65 | 28.81 |
| Cell Caspo EVs 2 | 3129623 | 3127440 | 97.49 | 6.79 | 61.69 | 31.52 |
| Cell Caspo EVs 3 | 2467369 | 2466065 | 94.58 | 9.23 | 49.35 | 41.42 |
| MMC1 sRNA | | | | | | |
| Cell 1 | 2756943 | 2753911 | 97.87 | 5.55 | 72.98 | 21.47 |
| Cell 2 | 2214280 | 2210472 | 96.43 | 4.78 | 77.31 | 17.91 |
| Cell 3 | 2029543 | 2026821 | 89.92 | 4.59 | 68.83 | 26.58 |
| Cell Caspo EVs 1 | 1905933 | 3441913 | 93.47 | 6.89 | 55.71 | 37.4 |
| Cell Caspo EVs 2 | 1931761 | 1900741 | 97.23 | 8.75 | 57.87 | 33.38 |
| Cell Caspo EVs 3 | 3464778 | 1930391 | 96.62 | 7.76 | 55.03 | 37.21 |
| Total sequences | 142'487'267 | 121'917'645 | | | | |
| Total nucleotides (nt) | 7,404,737,934 | 4,459,505,550 | | | | |

[a]Mapped to genes %: the percentage of reads that mapped to genes. Mapped to intergenic %: the percentage of reads that mappes to intergenic region. Mapped to ncRNAs %: percentage of reads that mapped to non coding RNAs. nt, nucleotides.

alignments: mismatch cost (2), insertion cost (3), deletion cost (3), length fraction (0.8), and similarity fraction (0.8). Only uniquely mapped reads were considered in the analysis. For mRNA identification in the EVs, we combined the differential expression with the read coverage, so we performed the map reads to reference (C_auris_B8441_version_s01-m01-r10_genomic and C_auris_B8441_version_s01-m01-r10_other_features_plus_intergenic) using the following parameters: no masking, match score (1), mismatch cost (2), linear insertion cost (3), deletion cost (3), length fraction (0.6), similarity fraction (0.8), and global alignment. For the validation of RNA-seq data, we performed differential gene expression analysis. The statistical test applied was the DGE (differential gene expression) test using the RNA-seq package with CLC Genomics Workbench v 20.0 (Qiagen). The expression values for the transcripts were registered in TPM (transcripts per million), and TMM (trimmed mean of M values) was used as a normalization method (44). The parameters to select the differentially expressed transcripts were a 3-fold change (>3 FC) and a false discovery rate (FDR) below or equal to 0.05. The sequences generated from the small RNA-seq cell and EV were trimmed to remove the internal adapter that might have been present in the shorter sequences. The TruSeq small RNA adapter sequence

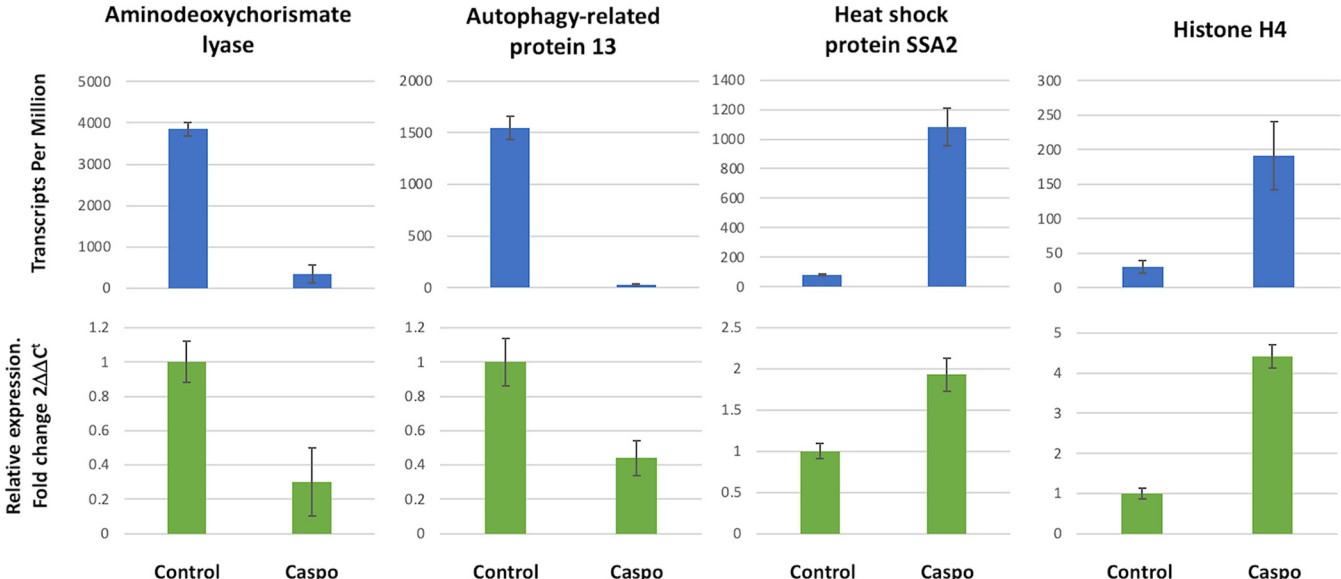

**FIG 7** Validation of RNA-seq data. RNA-seq expression levels indicated in TPM (transcripts per million) are shown in the top bar graphs. Error bars represent the mean and standard error of the biological replicates from all the strains used in this study ($n = 9$). For the qPCR, the relative expression of transcript levels was normalized to the corresponding level of the guanine nucleotide-binding protein transcript, and error bars represent the mean and standard error of quadruplicate samples performed twice from all the strains used in this study (bottom bar graphs). Reverse transcription-qPCR was performed according to MIQE guidelines.

was TGGAATTCTCGGGTGCCAAGG. When the adapter was identified in the sequence, it was removed along with the following sequence (3′ trim).

**Quantitative RT-PCR.** For the RNA-seq data validation, we used quantitative real-time PCR. The assay followed the Minimum Information for Publication of Quantitative Real-Time PCR Experiments (MIQE) guidelines (45). Total RNA was isolated in triplicates from $1 \times 10^7$ *C. auris* cells using the miRNeasy RNA isolation kit (Qiagen) with adaptations. A 1:1 volume of glass beads was added to the lysis buffer along with the yeast cells, and the mixture was subjected to 10 rounds of vortex agitation (each round, 1 min at 4°C) in order to disrupt the fungal cell wall. After centrifugation, total RNA was isolated according to the manufacturer's instructions and quantified using a Qubit fluorometer RNA HS kit (Thermo Fisher), and RNA integrity was assessed with a Bioanalyzer RNA PICO 6000 (Agilent). After isolation, 100 ng of cellular RNA and 20 ng of EV RNA was treated with 1 U of DNase I RNase-free (catalog no. EN0521 Promega) according to the manufacturer's instructions. After this, the cDNA was synthesized from 1 µg of cellular RNA or from 20 ng of vesicular RNA as templates. For the reverse transcriptase reactions 0.3 µM random primer (Invitrogen) and 1 µL of reverse transcriptase (Superscript II, Thermo Scientific), according to the manufacturers' instructions. PCR was performed with 40 ng of cDNA for the cell and 1.6 ng of cDNA for the EV as the template and GoTaq master mix according to the manufacturer's instructions (Promega). The oligonucleotides were designed with PRIMER-BLAST using the following parameters: PCR product size maximum of 250 nt, $T_m$ varying of 60°C, RefSeq mRNA as a database, and *Candida auris* as the organism. The primer sets used for PCR are described below. The qPCR was performed in three technical replicates for each sample. The following program was used in the Lightcycler 480 (Roche) equipment: initial denaturation at 95°C for 15 min, and 45 cycles of 95°C for 15 s, 60°C for 20 s, and 72°C for 45 s. The primers used in this study are listed, and the reference gene used was a guanine nucleotide-binding protein subunit beta-like protein (Table 3).

**TABLE 3** Primers used in this study

| Gene ID | Gene name | Direction | Sequence |
|---|---|---|---|
| B9J08_003083 | Aminodeoxychorismate lyase | F | CCTCAAAGCGTGGATGAGGT |
| | | R | CGCCGTCCAACTTGAGTAGT |
| B9J08_002073 | Histone H4 | F | TGGTCTTCTTCCCCATTGGC |
| | | R | GTCAAAAACGTCCAGGTGCC |
| B9J08_003364 | Heat shock protein SSA2 | F | GACCAAGACCTTCACCCCAG |
| | | R | AACCAGCATCCTTGGTAGCC |
| B9J08_005458 | Guanine nucleotide-binding protein subunit beta-like protein | F | CACCCCATCCAAGCCTGATT |
| | | R | AGGCACCGTCAACAGAGATG |
| B9J08_000925 | Autophagy-related protein 13 | F | AACTCTGCATTGGACTCCCG |
| | | R | ATGTTTCCTCGTGCTGCTCA |

**Data availability.** RNA-seq raw data (fastq files) related to this study were submitted to the NCBI repository SRA (SRA: SRP295539 BioProject: PRJNA682185). Mapping statistics are available (Table 2).

## ACKNOWLEDGMENTS

We thank the staff of the Genomics section of the Life Sciences Core Facility (LaCTAD), part of the University of Campinas (UNICAMP), for their contributions to RNA-sequencing.

J.D.N. and D.Z-M. were partially supported by NIH R21 AI124797. M.L.R. was supported by grants from the Brazilian Ministry of Health (grant no. 440015/2018-9), Conselho Nacional de Desenvolvimento Científico e Tecnológico (CNPq, grant no. 405520/2018-2 and no. 301304/2017-3) and Fiocruz (grant no. VPPCB-007-FIO-18 and no. VPPIS-001-FIO18). The authors also acknowledge support from the Instituto Nacional de Ciência e Tecnologia de Inovação em Doenças de Populações Negligenciadas (INCT-IDPN). M.L.R. is currently on leave from the position of Associate Professor at the Microbiology Institute of the Federal University of Rio de Janeiro, Brazil. L.R.A. received financial support from Inova Fiocruz/Fundação Oswaldo Cruz (grant no. VPPCB-07-FIO-18-2-52) and CNPq (grant no. 442317/2019-0). L.R.A is a research fellow awardee from CNPq.

I.F.M.R. performed the extracellular vesicles analysis. R.F.A. and D.Z-M. performed most of the experiments. S.T.M. and L.R.A. performed the RNA-seq analysis. M.L.R. and J.D.N. discussed all the results. I.F.M.R., R.F.A., D.Z-M., S.T.M., M.L.R., J.D.N., and L.R.A. helped to write the manuscript. All the authors approved the final version of the manuscript.

We declare no conflicts of interest.

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
