## [Reviewer comments · Microbiology Spectrum]

Microbiology Spectrum

Cellular and extracellular vesicle RNA analysis in the global threat fungus *Candida auris*

Isadora Munhoz da Rocha, Sharon Martins, Rafaela Amatuzzi, Daniel Zamith-Miranda, Joshua Nosanchuk, Marcio Rodrigues, and Lysangela Alves

Corresponding Author(s): Lysangela Alves, Instituto Carlos Chagas (ICC), Fundação Oswaldo Cruz

Review Timeline:

Submission Date:	September 9, 2021
Editorial Decision:	October 8, 2021
Revision Received:	October 26, 2021
Editorial Decision:	October 28, 2021
Revision Received:	October 29, 2021
Editorial Decision:	November 2, 2021
Revision Received:	November 2, 2021
Accepted:	November 3, 2021

Editor: Kirsten Nielsen

Reviewer(s): The reviewers have opted to remain anonymous.

Transaction Report:

DOI: <https://doi.org/10.1128/Spectrum.01538-21>

Dr. Lysangela Ronalte Alves
Instituto Carlos Chagas (ICC), Fundação Oswaldo Cruz
Laboratório de Regulação Gênica
Rua Algacyr Munhoz Mader, 3775
CIC
Curitiba, Paraná 81350010
Brazil

Re: Spectrum01538-21 (Cellular and extracellular vesicle RNA analysis in the global threat fungus *Candida auris*)

Dear Dr. Lysangela Ronalte Alves:

I have received the reviews of your manuscript entitled "Cellular and extracellular vesicle RNA analysis in the global threat fungus *Candida auris*", and I regret to inform you that we will not be able to publish it in Spectrum. Your submission was read by reviewers with expertise in the area addressed in your study and it was the consensus view of these reviewers that your paper did not meet the standards necessary for publication. Copies of the reviewers' comments are attached for your consideration.

Both reviewers felt that the manuscript was descriptive in nature and lacked a data-driven hypothesis. Both reviewers also felt the data analysis was very superficial and thus the current state of the manuscript was significantly below the publication criteria of Microbiology Spectrum.

I am sorry to convey a negative decision on this occasion, but I hope that the enclosed reviews are useful. Please note, rejections from Microbiology Spectrum are final and your manuscript will not be considered by other ASM journals. We wish you well in publishing this report in another journal and hope that you will consider Spectrum in the future.

Sincerely,

Kirsten Nielsen
Editor, Microbiology Spectrum

Reviewer comments:

Reviewer #1 (Comments for the Author):

Munhoz da Rocha and colleagues described their attempts in characterizing the secreted vesicular and cellular small RNAs from the three strains of *Candida auris* with- and without the presence of the lipopeptide antifungal drug Caspofungin. They described mainly the experimental procedures and parameters of their finding. Their effort in attempting to provide a dataset benefiting the research field is worthy of applauding; however, as the report's current strands, the effort would not be able to benefit anyone.

Main comments:

This is a transcriptomic study of secreted and cellular small RNAs from *C. auris*, but no data annotation, such as analysis of functional gene ontology (GO) terms or biological pathways (KEGG pathways), was performed. Without it, the study is not publishable.

The report listed figure and table numbers in the text but no data interpretation was provided. Most of the Results and Discussions is just the extended description of Materials and Methods.

Specific comments:

Lines 24-26: The preceding lines are statements on extracellular vesicles, but the sentences here do not follow through and suddenly turn into small cellular RNAs.

Lines 278-30: Need to be specific, such as "sRNA-mediated regulation of the gene expression in response to antifungals."

Line 75: why "viral infection" is mentioned here?

Line 79: "The proteins of RNA silencing pathway were lost in a significant number of fungal species." The statement makes no sense to be here and is not supported by reference (no reference was provided).

Lines 87-97: These statements are somewhat erroneous. Authors may wish to initiate a literature search to be up to date with the current study progress.

Reviewer #2 (Comments for the Author):

Dear Editor,

The manuscript by Rocha et al describes the composition and identity of the small RNA fraction isolated from cellular and extracellular vesicles (EVs) compartments from *Candida auris*. The authors describes a slight alteration in the EVs profile of three distinct *C. auris* strains exposed to the antifungal drug caspofungin. In addition, employing small RNA-seq analysis, the authors found that yeast cells treated with caspofungin displayed a substantial difference in the small RNA composition in EVs compared to cellular compartment, as revealed by PCA analysis. They also found that caspofungin itself caused an alteration in the small RNA profile in such strains.

While the manuscript is well written, it lacks proper analysis of the results and consequent data-dependent generation of hypothesis. There are some major points that are illustrative of my previous comment:

1 - Based on *Candida albicans* previous studies (<https://pubmed.ncbi.nlm.nih.gov/23114781/>), caspofungin leads to apoptosis in yeast cells. The increased EV content in B8441 and MMC1 strains could be potentially associated with apoptosis. This is not discussed in the manuscript and could be easily evaluated by the TUNEL assay. Moreover, in Figure 2, there is no statistical analysis to compare the groups. Despite described in the text, there is no images of the isolated EVs.

2 - The authors generated a large dataset of small RNA sequences from *C. Auris* that certainly will be useful to the community. However, the analysis is merely descriptive about the general aspects of size distribution and content of such libraries. The authors could use such data to infer what are the underlying mechanisms about the (i) distinct sensitivity of the strains to caspofungin and (ii) distinct profile of EVs produced by the strains. Moreover, the author could produce a description of distinct classes of sRNAs (miRNA like, tRNA derived fragments, etc), as well as their differential abundance in the treatments and the strains. In addition, the authors could compare their data to their previous publication (doi.org/10.1016/j.csbj.2021.09.007) in order to correlate the transcription profiling of mRNAs and sRNAs of *C. auris* in response to caspofungin.

Minor points

3 - Why the fraction of unmapped reads is so high?

4 - Based on which criteria the RIN was determined? There is some variation in the parameters according to the organism used.

5 - Why did the authors mapped the reads against two distinct libraries? The sentence that describes such information is not clear (lines 132-135). If this was made for annotation purposes, Why did not the authors merge such annotations in a single annotation file (gff or gtf) and process the alignments files? This would also filter reads spanning multiple aligned reads that were considered in the proposed approach.

6 - Lines 103 - 105. The authors stated that they conducted RNA-seq to evaluate the drug effect on living cells. Despite a clear signal could be observed in MIC assays, it is expected that some cells are dead. Does the authors evaluated the viability of the cells exposed to such drugs concentrations used for RNA seq analysis?

October 28, 2021

Dr. Lysangela Ronalte Alves
Instituto Carlos Chagas (ICC), Fundação Oswaldo Cruz
Laboratório de Regulação Gênica
Rua Algacyr Munhoz Mader, 3775
CIC
Curitiba, Paraná 81350010
Brazil

Re: Spectrum01538-21R1-A (Cellular and extracellular vesicle RNA analysis in the global threat fungus *Candida auris*)

Dear Dr. Lysangela Ronalte Alves:

I'm sorry for the previous confusion. As the editor, I had missed the important notation that your manuscript was submitted as a "Resource Report" and not a "Research Article". For this I sincerely apologize and I am glad that you appealed my initial decision. In your response to the reviewer comments, please note that your manuscript is a resource that is being provided to the research community. I would also recommend that you include additional language in the abstract, introduction, and discussion to highlight to the research community that these data are being provided as a resource since this is a new format for our community and will prevent confusion such as occurred here in the future when your paper is published. Again, I'm very sorry for the mistake on my part.

Thank you for submitting your manuscript to Microbiology Spectrum. When submitting the revised version of your paper, please provide (1) point-by-point responses to the issues raised by the reviewers as file type "Response to Reviewers," not in your cover letter, and (2) a PDF file that indicates the changes from the original submission (by highlighting or underlining the changes) as file type "Marked Up Manuscript - For Review Only". Please use this link to submit your revised manuscript - we strongly recommend that you submit your paper within the next 60 days or reach out to me. Detailed information on submitting your revised paper are below.

Link Not Available

Sincerely,

Kirsten Nielsen

Journals Department
Reviewer comments:

Reviewer #1:

Munhoz da Rocha and colleagues described their attempts in characterizing the secreted vesicular and cellular small RNAs from the three strains of *Candida auris* with- and without the presence of the lipopeptide antifungal drug Caspofungin. They described mainly the experimental procedures and parameters of their finding. Their effort in attempting to provide a dataset benefiting the research field is worthy of applauding; however, as the report's current strands, the effort would not be able to benefit anyone.

Main comments:

This is a transcriptomic study of secreted and cellular small RNAs from *C. auris*, but no data annotation, such as analysis of functional gene ontology (GO) terms or biological pathways (KEGG pathways), was performed. Without it, the study is not publishable.

The report listed figure and table numbers in the text but no data interpretation was provided. Most of the Results and Discussions is just the extended description of Materials and Methods.

Specific comments:

Lines 24-26: The preceding lines are statements on extracellular vesicles, but the sentences here do not follow through and suddenly turn into small cellular RNAs.

Lines 278-30: Need to be specific, such as "sRNA-mediated regulation of the gene expression in response to antifungals."

Line 75: why "viral infection" is mentioned here?

Line 79: "The proteins of RNA silencing pathway were lost in a significant number of fungal species." The statement makes no sense to be here and is not supported by reference (no reference was provided).

Lines 87-97: These statements are somewhat erroneous. Authors may wish to initiate a literature search to be up to date with the current study progress.

Reviewer #2:

The manuscript by Rocha et al describes the composition and identity of the small RNA fraction isolated from cellular and extracellular vesicles (EVs) compartments from *Candida auris*. The authors describes a slight alteration in the EVs profile of three distinct *C. auris* strains exposed to the antifungal drug caspofungin. In addition, employing small RNA-seq analysis, the authors found that yeast cells treated with caspofungin displayed a substantial difference in the small RNA composition in EVs compared to cellular compartment, as revealed by PCA analysis. They also found that caspofungin itself caused an alteration in the small RNA profile in such strains.

While the manuscript is well written, it lacks proper analysis of the results and consequent data-dependent generation of hypothesis. There are some major points that are illustrative of my previous comment:

1 - Based on *Candida albicans* previous studies (<https://pubmed.ncbi.nlm.nih.gov/23114781/>), caspofungin leads to apoptosis in yeast cells. The increased EV content in B8441 and MMC1 strains could be potentially associated with apoptosis. This is not discussed in the manuscript and could be easily evaluated by the TUNEL assay. Moreover, in Figure 2, there is no statistical analysis to compare the groups. Despite described in the text, there is no images of the isolated EVs.

2 - The authors generated a large dataset of small RNA sequences from *C. Auris* that certainly will be useful to the community. However, the analysis is merely descriptive about the general aspects of size distribution and content of such libraries. The authors could use such data to infer what are the underlying mechanisms about the (i) distinct sensitivity of the strains to caspofungin and (ii) distinct profile of EVs produced by the strains. Moreover, the author could produce a description of distinct classes of sRNAs (miRNA like, tRNA derived fragments, etc), as well as their differential abundance in the treatments and the strains. In addition, the authors could compare their data to their previous publication (doi.org/10.1016/j.csbj.2021.09.007) in order to correlate the transcription profiling of mRNAs and sRNAs of *C. auris* in response to caspofungin.

Minor points

3 - Why the fraction of unmapped reads is so high?

4 - Based on which criteria the RIN was determined? There is some variation in the parameters according to the organism used.

5 - Why did the authors mapped the reads against two distinct libraries? The sentence that describes such information is not clear (lines 132-135). If this was made for annotation purposes, Why did not the authors merge such annotations in a single annotation file (gff or gtf) and process the alignments files? This would also filter reads spanning multiple aligned reads that were considered in the proposed approach.

6 - Lines 103 - 105. The authors stated that they conducted RNA-seq to evaluate the drug effect on living cells. Despite a clear signal could be observed in MIC assays, it is expected that some cells are dead. Does the authors evaluated the viability of the cells exposed to such drugs concentrations used for RNA seq analysis?

Staff Comments:

Preparing Revision Guidelines

Please return the manuscript within 60 days; if you cannot complete the modification within this time period, please contact me. If you do not wish to modify the manuscript and prefer to submit it to another journal, please notify me of your decision immediately so that the manuscript may be formally withdrawn from consideration by Microbiology Spectrum.

November 2, 2021

Dr. Lysangela Ronalte Alves
Instituto Carlos Chagas (ICC), Fundação Oswaldo Cruz
Laboratório de Regulação Gênica
Rua Algacyr Munhoz Mader, 3775
CIC
Curitiba, Paraná 81350010
Brazil

Re: Spectrum01538-21R2 (Cellular and extracellular vesicle RNA analysis in the global threat fungus *Candida auris*)

Dear Dr. Lysangela Ronalte Alves:

The authors appropriately responded to the reviewers comments. I requested modifications because the manuscript includes several areas where the authors still include their "(ref)" annotation without the appropriate reference included. Assuming the authors replace these annotations with appropriate citations, the manuscript should be acceptable for publication.

Thank you for submitting your manuscript to Microbiology Spectrum. As you will see your paper is very close to acceptance. Please modify the manuscript along the lines I have recommended. As these revisions are quite minor, I expect that you should be able to turn in the revised paper in less than 30 days, if not sooner. If your manuscript was reviewed, you will find the reviewers' comments below.

When submitting the revised version of your paper, please provide (1) point-by-point responses to the issues I raised in your cover letter, and (2) a PDF file that indicates the changes from the original submission (by highlighting or underlining the changes) as file type "Marked Up Manuscript - For Review Only". Please use this link to submit your revised manuscript. Detailed information on submitting your revised paper are below.

Link Not Available

Sincerely,

Kirsten Nielsen

Reviewer comments:

Preparing Revision Guidelines

- point-by-point responses to the issues I raised in your cover letter
- Upload a compare copy of the manuscript (without figures) as a "Marked-Up Manuscript" file.
- Each figure must be uploaded as a separate file, and any multipanel figures must be assembled into one file.
- Manuscript: A .DOC version of the revised manuscript
- Figures: Editable, high-resolution, individual figure files are required at revision, TIFF or EPS files are preferred

Please return the manuscript within 60 days; if you cannot complete the modification within this time period, please contact me. If you do not wish to modify the manuscript and prefer to submit it to another journal, please notify me of your decision immediately so that the manuscript may be formally withdrawn from consideration by Microbiology Spectrum.

Ministério da Saúde

FIOCRUZ - PARANÁ
Instituto Carlos Chagas

Dear Dr. Nielsen,

We really appreciate that you allowed us to resubmit the manuscript “Cellular and extracellular vesicle RNA analysis in the global threat fungus *Candida auris*” to Microbiology Spectrum.

We thank you again for all the efforts in order to improve our manuscript, we apologize for missing the references in the text, they were appropriately included (lines 171 and 174).

Sincerely,

Instituto Carlos Chagas Fiocruz-PR, Curitiba, PR, Brazil

lysangela.alves@fiocruz.br, lys.alves@gmail.com

+55 41 3316 3230

+55(41)3316-3230

Rua Prof. Algacyr Munhoz Mader, 3775 - CIC 81350-010 Curitiba/PR, Brasil

November 3, 2021

Dr. Lysangela Ronalte Alves
Instituto Carlos Chagas (ICC), Fundação Oswaldo Cruz
Laboratório de Regulação Gênica
Rua Algacyr Munhoz Mader, 3775
CIC
Curitiba, Paraná 81350010
Brazil

Re: Spectrum01538-21R3 (Cellular and extracellular vesicle RNA analysis in the global threat fungus *Candida auris*)

Dear Dr. Lysangela Ronalte Alves:

Your manuscript has been accepted, and I am forwarding it to the ASM Journals Department for publication. You will be notified when your proofs are ready to be viewed.

Sincerely,

Kirsten Nielsen
Editor, Microbiology Spectrum
